# Understanding Interpersonal Influences on Maternal Health Service Utilization at Community Health Centers: A Mixed-Methods Study in Indonesia

**DOI:** 10.3390/healthcare13010042

**Published:** 2024-12-30

**Authors:** Herwansyah Herwansyah, Katarzyna Czabanowska, Stavroula Kalaitzi, Peter Schröder-Bäck

**Affiliations:** 1Department of International Health, Care and Public Health Research Institute (CAPHRI), Faculty of Health, Medicine and Life Sciences, Maastricht University, P.O. Box 616, 6200 MD Maastricht, The Netherlands; 2Public Health Study Program, Faculty of Medicine and Health Sciences, Universitas Jambi, Telanaipura, Kota Jambi 36122, Indonesia; 3Department of Health Policy and Management, Institute of Public Health, Faculty of Health Sciences, Jagiellonian University, 31-007 Krakow, Poland; 4Department of Educational Studies, National and Kapodistrian University of Athens, 115 28 Athens, Greece; 5Institute of History and Ethics of Police and Public Administration (IGE), University of Applied Sciences for Police and Public Administration, 52068 Aachen, Germany

**Keywords:** interpersonal support, maternal health services, community health center, mixed methods

## Abstract

**Background/Objective**: The utilization of maternal health services at the primary healthcare level is still considered an effective approach despite the critical role in improving maternal health outcomes. The study aimed to assess the influence of sociodemographic characteristics and interpersonal support on the use of maternal health services in three regions of the Province of Jambi, Indonesia. **Methods**: Using a mixed-methods sequential explanatory design, a quantitative survey of 432 women and qualitative focus group discussions with nine families were conducted. Quantitative data were analyzed using bivariate analysis, while the qualitative data were examined through conventional content analysis. **Results**: The research results show no significant association between sociodemographic factors (education, employment, residence) and the use of maternal health services for antenatal care and delivery. Direct support, such as accompaniment by spouses, and indirect support, including emotional encouragement and shared knowledge, were pivotal in influencing women’s decisions to seek care. **Conclusions**: Targeted interventions focusing on improving social support, addressing service accessibility barriers, and raising awareness about the benefits of community health centers are essential to enhancing maternal health outcomes. Policymakers and healthcare providers should integrate family-centered strategies to ensure women receive adequate maternal healthcare.

## 1. Background

Maternal mortality is a multifaceted global public health issue that requires comprehensive efforts from all elements, including governments, healthcare providers, and education sectors [1,2]. The concerted attempts aim to improve the quality of women’s health and address the underlying social determinants that contribute to high global maternal deaths [3]. Maternal mortality is considered one of the essential indicators of overall health and well-being that reflects the quality of healthcare services in a given country [4]. Therefore, sustained and prompt actions must be taken to achieve one of the key components of sustainable development worldwide.

The high maternal mortality ratio remains a public health concern in Indonesia and has been a focus of the national program for many years [5,6]. Recent data indicate that the Maternal Mortality Ratio (MMR) in Indonesia remains one of the highest in Southeast Asia, accounting for 173 per 100,000 live births, highlighting persistent challenges in ensuring access to adequate maternal health services [7]. Several programs, including the optimization of maternal health service utilization at the primary healthcare level, have been implemented widely. A community health center (CHC) is essential for achieving better maternal health status since it has been equipped with an adequate number of health workers and basic facilities for maternity care and is easy to access. CHCs serve as the primary providers of maternal health services in Indonesia, offering antenatal care, skilled delivery assistance, postnatal care, and maternal health education to ensure the well-being of mothers and newborns [8]. CHC is considered the first entry point into the entire healthcare system, which enables a broad range of health services provided by health professionals to support the community’s health needs [9]. However, the utilization rate of maternal health services at the CHC is lower than expected. This phenomenon indicates that the population prefers to use personal pocket money to pay for the services from private healthcare facilities [10].

Studies have identified various factors influencing the utilization of maternal health services at the CHCs [11,12,13,14]. These factors include the characteristics of the mother and family, family support, healthcare providers, and other social determinants. Although the family may seem to play a minor role in society, it has a significant impact on the lives of individuals, especially those facing challenges. Family encouragement is particularly essential in motivating women to access maternal health services at CHCs, which can significantly enhance the possibility of achieving positive health outcomes. In addition, as in many countries in Southeast Asia, the family is considered a key decision-making unit in determining access to maternal health services [15,16].

It is essential to understand the influential people in the family in order to develop strategies and set goals to promote maternal health service utilization at the community level. To the best of our knowledge, no recent studies have specifically examined the interpersonal factors associated with maternal health service utilization at the CHC in the Indonesian context. This study aimed to explore household perspectives and assess the socio-demographic characteristics of household members in light of their support for the use of maternal health services.

## 2. Methods

### 2.1. Research Design and Settings

This study is based on a mixed-methods explanatory sequential design [17,18] (Figure 1) to obtain a more holistic understanding of interpersonal determinants associated with maternal health service utilization. The quantitative approach accounts for the main research, which was conducted through a survey of 432 respondents concerning the characteristics of women and their husbands as well as maternal health-seeking behavior. The qualitative research examined nine families’ perceptions of support of maternal health service utilization and the actions they took to address barriers to service utilization from the interpersonal level. The study was conducted in three regencies of Jambi Province (Municipality of Jambi, Municipality of Sungai Penuh, and Merangin Regency). Some considerations of selecting study settings include (1) representing western, central, and eastern regions; (2) three regions have a significant difference in maternal mortality; and (3) all CHCs are accessible.

### 2.2. Sampling Process

The study population of this research included reproductive-aged women who had given birth at least once and resided in the area for the last five years. Data on reproductive-aged women were obtained from the provincial health department. The number of respondents required for the survey was calculated using Slovin’s formula [19]:
n = N/(1 + N e^2^)
where N is the expected population size (347,463), and e is the error tolerance (0.05). This study aims at a minimum target sample size of 384 respondents based on the formula. Additional respondents (10% of the minimum target sample size) were included in this study due to the possibility of selective non-response and dropout, as well as to ensure sufficient respondents at the various CHCs. The final sample size was estimated to be 423 respondents. Convenience sampling was employed to provide flexibility in reaching out to potential informants of the qualitative data [20]. Nine families were interviewed in three regions. The interviewees included the wives, husbands, parents, and other family members who reside in the same household.

### 2.3. Quantitative Phase

The quantitative survey was developed to elucidate the characteristics of the targeted respondents. Prior to the distribution of the questionnaire, a pilot study was conducted with 30 participants to refine the questions and ensure clarity, relevance, and reliability. This pilot testing process was crucial for identifying potential issues with question phrasing, improving the overall design of the survey, and mitigating the risk of information bias in the final data collection. The outcome variables included willingness to visit the CHC for the first ANC, the subsequent ANC visit, and utilization of the delivery services at the CHC. Independent variables comprised the socio-demographic characteristics of the woman, her husband, and the family’s residency status.

We conducted in-person interviews with 432 women who came to the community health centers and resided at the research sites. The first authors led the data collection with assistance from research assistants. The survey dataset was calculated to generate descriptive data and followed by a bivariate analysis to determine the association between variables. The dataset was initially processed to generate descriptive statistics, which provided an overview of the characteristics of the respondents, including age, employment status, education level, and proximity to CHCs. These descriptive statistics offered insights into the sociodemographic profile of the study population. Following this, a bivariate analysis was conducted to determine the relationships between the dependent variable (maternal health service utilization) and various independent variables (such as sociodemographic characteristics, proximity to health centers, and family support). Logistic regression was utilized to assess the relative importance of various predictors and identify the most relevant factors associated with maternal health service utilization. Data were analyzed using SPSS (IBM SPSS version 24).

### 2.4. Qualitative Phase

Another essential step in this study was the qualitative research phase, where we collected valuable information to investigate and comprehend social phenomena from Focus Group Discussions (FGDs) with family members. Participants for the FGDs were purposively selected with the assistance of the CHC staff, who identified individuals based on their involvement with maternal health decision-making in the family. The selection of diverse family members aimed to capture a diverse range of perspectives on maternal health service utilization.

The following main questions were discussed with the participants, with the prompt changing depending on the situation during the discussion: (1) what do your family members know about the services available at community health centers for maternal health? (2) based on experience, how has the family supported or influenced the women’s decisions regarding maternal health service utilization? (3) are there any concerns or fears within your family that might hinder your use of maternal health services?

The purpose of the questions was to generate a deeper understanding of how individuals with interpersonal relationships with women perceive their role in supporting maternal healthcare service utilization at the CHCs. Data were collected in a comfortable and informal setting. The first author led the discussion using the most familiar language for the participants. Follow-up prompts and probing questions were used to clarify responses and gain deeper insights into participants’ views on the topic. FGDs were audio recorded and transcribed into Bahasa Indonesia. The first author coded each transcript and performed a conventional content analysis [21] in which codes were derived from the larger data in the transcripts.

### 2.5. Ethical Considerations

The main study protocol was approved by the Ethical Committee of the Faculty of Medicine and Health Sciences Universitas Jambi, Indonesia, before commencing data collection (Ref. no 795/UN21.8/PT.01.04/2021). Each regency’s Department of Health granted permission for data collection at the participating research locations. All participants provided their written informed consent to participate in the study after reading a statement outlining the study’s objectives.

## 3. Results

### 3.1. Socio-Demographics

Table 1 presents the socio-demographic profile of the respondents who participated in the study. More than 50% of the respondents were aged between 21–30 years. A great majority (more than 60%) was employed and had jobs. The Malay tribe made up the majority of the respondents in this study, accounting for more than 40% of the total respondents. The proportion of respondents with high and low education differed marginally between women and their husbands. About half of the sample (53.0%) lived with the extended family. Additionally, a significant proportion (71.4%) of the respondents resided close to community health centers.

### 3.2. Maternal Health Service Utilization

Table 2 shows the characteristics of maternal health service utilization at the CHC by the respondents. Almost 65% of the respondents reported that they had their first ANC visit at the CHC. The majority of respondents (61.7%) visited other healthcare facilities other than the CHC for their subsequent ANC. Furthermore, the largest group (73.3%) of pregnant women had delivered at non-CHC healthcare facilities.

### 3.3. Bivariate Analysis

Table 3 shows that the group of women with low levels of education preferred community health centers (33.6%) as healthcare facilities to receive initial ANC services. However, the analysis result shows no significant association between women’s education level and their first ANC visit preference (*p* = 0.370). In addition, women with a low level of education used healthcare facilities other than CHCs for the upcoming ANC visits (34.0%) and delivery services (39.0%). Research shows that there is no significant correlation between women’s education level and the subsequent ANC visits or place of delivery reference, with respective values of *p* = 0.361 and *p* = 0.890.

Furthermore, the majority of working women (43.0%) received their first ANC services at a community health center. This also occurred at subsequent ANC visits and deliveries. Working women opted to visit healthcare facilities other than the CHCs for the following ANC visits and delivery services, with percentages of 38.5% and 47.5%, respectively. Research shows that there is no significant association between the employment status of the women and the decision for first ANC at the community health center, subsequent ANC, and delivery at a health facility other than the community health center with a *p*-value > 0.05.

The study further found other non-significant associations of spouse factors for women’s use of CHC, including first ANC visits, next ANC visits, and delivery services (*p* > 0.05). The majority of working spouses (60.8%) supported that their wives should receive initial ANC services from the CHCs, and the working husbands agreed to support their spouses to have subsequent ANC and delivery services from non-CHCs. The characteristics of husbands with low and high levels of education vary slightly across the wives’ decisions to use maternal health services.

Women who stayed with other family members visited the CHCs for their initial ANCs (34.5%). A significant number of the respondents from the women cohort received the next ANC (33.1%) and delivery services (37.8%) from other non-CHC healthcare facilities. The statistical test established that there was no significant relationship (*p* > 0.05) between women’s preference for maternity health service facilities and their residential status. The survey also discovered that nearly 50% of women who resided around CHCs had their first ANC there. On the other hand, less than 30% of the respondents used the maternal health services from CHCs for their subsequent ANCs and delivery services. A chi-square test shows a significant association between the distance to the CHCs and delivery service preferences (*p*-value = 0.019).

### 3.4. Content Analysis

We investigated the sociocultural factors contributing to the decision-making in utilizing maternal health services at the CHC for women during pregnancy, delivery, and post-partum care. The broad themes derived from the conventional content analysis are presented with the selected quotes below.

#### 3.4.1. Sufficient Understanding of the Importance of Maternal Health Service Utilization

The study found that the family members recognized the importance of maternal health service utilization provided by the CHCs. The use of these services is crucial for a number of reasons, including the following statement:


*“I gain knowledge about maternal health services at the CHC from the practical experiences of family members who work as healthcare professionals at the institution. I do agree with the information given by the family members. It is very important to have adequate services during pregnancy, delivery, and postpartum care. I also hear that the services provided are generally free as they are covered by the health insurance scheme of the government.”*
(F006)

An adequate understanding of how important it is to have comprehensive maternal health was described by one of the family members, who thought that there are varied maternal health programs that are only available at the CHCs. A family member shared the views in this regard:


*“I am aware of the significance of CHC based on my prior experience. There are numerous maternal health programs available. One of the most popular programs is called “Posyandu”, which refers to an integrated health service delivery post in the community. Apart from maternal health service provision, this service reflects community empowerment since the activity is run by the community, for the community, and facilitated by healthcare professionals from the CHCs.”*
(F007)

Families realized how important the services provided by primary healthcare facilities. Hence, they supported women in receiving appropriate maternal healthcare services. In addition, the families agreed that this is a crucial step in preserving the overall health and well-being of the family members; one of the family members stated:


*“Personally, I do believe that having adequate services from the nearest healthcare facilities is beneficial for everyone, including for women. If a woman in the family is in healthy, she will ensure the condition of other family members. Therefore, other family members should support the woman’s health, particularly their condition during pregnancy, delivery, and postpartum.”*
(F008)

#### 3.4.2. Family Support of Maternal Health Service Utilization

Support for women to utilize maternal health services at the CHC emerges when those around them recognize the value of the service. Both direct and indirect assistance, especially from the closest family members, is needed. Being a husband who is always ready to accompany the spouse to the CHC is one approach to demonstrating unconditional support. One informant reported the following:


*“Yes, I have to be a supportive spouse who is always available for my pregnant wife anytime she needs assistance to visit the CHC. Although we live with our parents, we do not want to bother them to accompany my wife for her monthly maternity appointment. I often have to schedule my work off on that date, so I can join my wife.”*
(F002)

Indirect support for women involves forms of assistance and efforts that are not directly related to physical aspects but still play an important role in providing holistic support. Women who are experiencing uncertainty during their pregnancy or after delivery can benefit from emotional support. Providing space for them to talk about their feelings is considered a very effective form of support. One of the family members revealed the following:


*“It seems that I do not contribute directly to my daughter’s support for using the maternal health service at the CHC. However, I always convince my daughter of the importance of maternal health services for her. In other words, I encourage her to use the services in order to understand about changes in her health during pregnancy and delivery. Another form of support is providing emotional support during pregnancy and after birth. And I provide a space for her to talk about her feelings and experiences.”*
(F001)

Spouses have an important role in offering support and ensuring that maternity care during pregnancy, delivery, and postpartum goes well. The roles include assisting the partners in finding reliable sources of information about pregnancy, delivery, and postnatal care. In addition, spouses collaborate with wives in making decisions regarding maternity care that suit the wives’ conditions. One of the informants shared the experience:


*“I am a working husband, but I do support my wife to have adequate maternal health service from the CHC. I remind her that it is time for ANC. We often discuss and talk about the further plan with regard to having maternity care at primary healthcare facilities. So we can decide together what the best option is for my wife.”*
(F005)

#### 3.4.3. Barriers to Providing Support for Maternal Health Services Use at the CHC

Several challenges hinder the provision of adequate support to women in utilizing maternal health services at the CHC. Daily activities make it challenging to offer women who want to access maternal health services at the CHC physical or emotional assistance, especially for those with demanding job schedules. One informant said:


*“As I explained previously, our house is very close to a private health facility. I have to work from 7 a.m. and finish at 5 p.m. It is nearly impossible to accompany my wife to use maternal health services at the CHC. We visit the private clinics in our residential area, and most importantly, they are open in the afternoon. So, I can obviously take my wife there since I have a flexible time after office hour.”*
(F004)

Another factor influencing women’s preference for alternative healthcare facilities over the CHC for maternal health services is distance from the CHC. The private clinics are available close to the area where they reside, with flexible service hours. One of the family members commented the following:


*“Our home is quite a distance from the CHC location. Additionally, just to get my wife to the CHC during work hours, I have to travel back and forth from my office to our house. It also takes time. We therefore made the decision to visit the private clinics that are close to our home. Since we have no other choices, we choose not to use the CHC’s for particular maternal health services.”*
(F006)

Some people may not understand the importance of maternal health services. This lack of understanding prevents them from seeking health support. A lack of information from other family members about available services can also be a barrier for women to utilize maternal health services at the CHC. One informant described the following:


*“Since we live only as wife and husband in this house, other family members do not have a direct influence on us. However, they often offered suggestions from their previous experiences. The suggestions sometimes are not relevant to the current situation, for instance, the use of traditional treatment for pregnant women. They believe that this is a culture that has been inherited from the family. In response to this, we simply accepted and applied what was best for us.”*
(F009)

## 4. Discussion

In this study, we explore the role and responsibilities of interpersonal relationships in women’s access to maternal health services at the primary healthcare level. A combined quantitative and qualitative approach provides a comprehensive insight into the association between husband and wife characteristics and the utilization of maternal health services at the CHC. This study also raises concerns about the ways in which families provide support for women to use these services.

The current quantitative study has revealed that the majority of women of diverse characteristics visited the CHC to receive comprehensive ANC services. Positive knowledge and attitude toward the importance of having adequate initial ANC led women to utilize the service at the CHC. The knowledge was mainly obtained from other family members who work at the CHC and the experiences of other women who visited the CHC for their first ANC service. The previous study conducted in Ghana [22] and Ethiopia [23] found that knowledge and ANC activities at primary healthcare facilities were positively correlated. In addition, the knowledge and willingness to visit the CHC demonstrated by the respondents to this research are supported by government policy. The regulation contains a mandatory first ANC visit and records the visit in a maternal and child health book that is obtained during the first pregnancy check-up at the CHC. The health center offers a number of supporting examinations, which are completely free of charge and not provided at other healthcare facilities [24]. Therefore, women prefer to receive their first integrated ANC services through the CHC over others.

Qualitative findings describe that women preferred to use healthcare facilities other than the CHCs in order to receive adequate subsequent ANC and delivery services. The results of this study indicate that the majority of the working women cohort opted to utilize private clinics for their maternity care. The working spouses supported their decision to visit non-CHC facilities. The main reason underlying this condition is related to the employment status of the partners and the service hours of the CHC. Working women found it difficult to modify their work schedules to accommodate the schedule of follow-up ANC visits at the CHC. The CHC service hours start in the morning and end in the early afternoon. Meanwhile, other healthcare facilities have flexible service hours. To address the specific challenges faced by working women, policymakers might consider strategies to improve the flexibility of CHC service hours or provide additional support, such as offering after-hours care or expanding mobile health services. A previous study revealed that most working women used ANC services at healthcare facilities in a low proportion [25]. Another study conducted in Bangladesh found that there was no significant association between women’s employment status and the number of ANC visits at the CHCs [26]. Working women’s decision to obtain advanced ANC services and working spouses’ support are influenced by various conditions, including working hours and the availability of flexible service hours from other healthcare facilities.

The quantitative and qualitative findings show consistency in highlighting the importance of knowledge, government policy, and service availability in influencing women’s access to maternal health services. The quantitative data suggest that most women prefer CHCs for initial ANC visits due to their free services, while the qualitative data reveal that working women seek private clinics for subsequent services due to scheduling conflicts with CHC hours. Both sets of findings emphasize the role of family support, particularly from spouses, in shaping women’s decisions regarding healthcare access.

The results further suggest a negative association between residential status and maternal health service use. The residential status of women refers to whether they live with a small family or with other extended families. Despite living with other family members, the women had prior information on the importance of maternal health service utilization at the CHC, which has been embedded in their mindset. In contrast to previous studies from India and Rwanda, which have shown that family support affected the quality of maternal healthcare use [25,27], women with poor support were at risk of a low attendance rate in maternal healthcare service utilization. Moreover, this current study found that women received means of support from other family members in utilizing maternal health services. The support included emotional support given to the women and the provision of an in-depth understanding of the importance of the services at healthcare facilities. The need for adequate information and the literature consistently indicates the importance of social support to mitigate the unexpected risks associated with maternal health outcomes and improve maternal health status [28,29].

The research limitation is that this is not a representative sample of all women and families in the province of Jambi. However, the survey was conducted in diverse demographic areas and employed a mixed-methods approach. The limited generalizability of the findings to other regions in Indonesia or similar settings is due to the specific cultural and demographic contexts of the study sites. Further study is required to involve a larger sample size and research sites, expanding the study to other regions in Indonesia to enhance generalizability. In addition, perspectives from healthcare providers and community leaders are also needed to enrich the research data.

## 5. Conclusions

In summary, this study emphasizes the critical role of interpersonal factors in the utilization of maternal health services at community health centers (CHCs) in Indonesia. The findings suggest that knowledge, positive attitudes from family members, and previous experiences significantly influence women’s decisions to seek comprehensive maternal health services at the primary healthcare level. In contrast, sociodemographic characteristics appear to have a lesser impact on service utilization. By understanding these interpersonal dynamics, healthcare providers and policymakers can develop more targeted interventions that focus on strengthening family support and increasing awareness about the importance of maternal health services. These efforts could ultimately improve maternal health service utilization and enhance the overall well-being of women across Indonesia.

## Figures and Tables

**Figure 1 healthcare-13-00042-f001:**
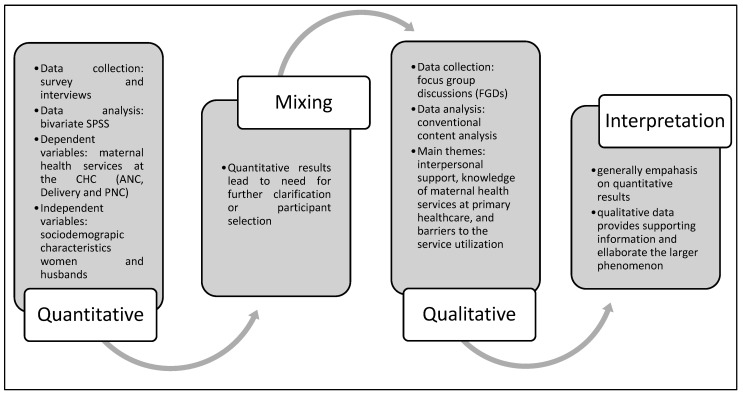
Explanatory sequential design in a mixed-methods study.

**Table 1 healthcare-13-00042-t001:** Socio-demographic characteristics for 423 respondents.

Characteristic	Women (N = 423)	Husbands (N = 423)	Both (N = 423)
Age			
≤20 years	21 (5%)	6 (1.4%)	-
21–30 years	321 (75.9%)	227 (53.7%)	-
≥31 years	81 (19.1%)	190 (44.9%)	-
	423 (100%)	423 (100%)	-
Employment			
Work	268 (63.4%)	399 (94.3%)	-
Not work	155 (36.6%)	24 (5.7%)	-
	423 (100%)	423 (100%)	-
Tribe			
Malay	193 (45.6%)	189 (44.7%)	-
Minangnese	67 (15.8%)	68 (16.1%)	-
Bataknese	37 (8.7%)	49 (11.6%)	-
Javanese	82 (19.4%)	74 (17.5%)	-
others	44 (10.4%)	43 (10.2%)	-
	423 (100%)	423 (100%)	-
Education			
Low	226 (53.4%)	202 (47.8%)	-
High	197 (46.6%)	221 (52.2%)	-
	423 (100%)	423 (100%)	-
Residential characteristics			
Extended family	-	-	224 (53.0%)
Nuclear family	-	-	199 (47.0%)
	-	-	423 (100%)
Distance to CHC			
Near	-	-	302 (71.4%)
Far	-	-	121 (28.6%)
	-	-	423 (100%)

**Table 2 healthcare-13-00042-t002:** Characteristics for maternal health service utilization.

Characteristic	N (%)
First ANC at CHC	
Yes	274 (64.8%)
No	149 (35.2%)
	423 (100%)
Next ANC visit	
Non CHC	261 (61.7%)
CHC	162 (38.3%)
	423 (100%)
Delivery	
Non CHC	310 (73.3%)
CHC	113 (26.7%)
	423 (100%)

**Table 3 healthcare-13-00042-t003:** Comparison of women’s and husband’s socio-demographic characteristics of maternal health service utilization.

Characteristics	First ANC at CHC		*p*-Value	Next ANC Visit		*p*-Value	Delivery		*p*-Value
	Yes	No		Non CHC	CHC		Non CHC	CHC	
Women’s Education			0.370			0.361			0.890
Low	142 (33.6%)	84 (19.9%)		144 (34.0%)	82 (19.4%)		165 (39.0%)	61 (14.4%)	
High	132 (31.2%)	65 (15.4%)		117 (27.7%)	80 (18.9%)		145 (34.3%)	52 (12.3%)	
Women’s Employment			0.076			0.624			0.295
Work	182 (43.0%)	86 (20.3%)		163 (38.5%)	105 (24.8%)		201 (47.5%)	67 (15.8%)	
Not work	92 (21.7%	63 (14.9%)		98 (23.2%)	57 (13.5%)		109 (25.8%)	46 (10.9%)	
Husband’s Education			0.814			0.466			0.819
Low	132 (31.2%)	70 (16.5%)		121 (28.6%)	81 (19.1%)		147 (34.8%)	55 (13.0%)	
High	142 (33.6%)	79 (18.7%)		140 (33.1%)	81 (19.1%)		163 (38.5%)	58 (13.7%)	
Husbands’ Employment			0.522			0.434			0.780
Work	257 (60.8%)	142 (33.6%)		248 (58.6%)	151 (35.7%)		293 (69.3%)	106 (25.1%)	
Not work	17 (4.0%)	7 (1.7%)		13 (3.1%)	11 (2.6%)		17 (4.0%)	7 (1.7%)	
Residential characteristics			0.854			0.720			0.360
Extended family	146 (34.5%)	78 (18.4%)		140 (33.1%)	84 (19.9%)		160 (37.8%)	64 (15.1%)	
Nuclear family	128 (30.3%)	71 (16.8%)		121 (28.6%)	78 (18.4%)		150 (35.5%)	49 (11.6%)	
Distance to CHC			0.889			0.140			0.019
Near	195 (46.1%)	107 (25.3%)		193 (45.6%)	109 (25.8%)		231 (54.6%)	71 (16.8%)	
Far	79 (18.7%)	42 (9.9%)		68 (16.1%)	53 (12.5%)		79 (18.7%)	42 (9.9%)	

## Data Availability

The data presented in this study are available on request from the corresponding author.

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
