# Peer review of "Understanding Interpersonal Influences on Maternal Health Service Utilization at Community Health Centers: A Mixed-Methods Study in Indonesia"

_healthcare, 2024, doi:10.3390/healthcare13010042_

Round 1
Reviewer 1 Report
Comments and Suggestions for Authors
Dear Authors
The article "Understanding Interpersonal Influences on Maternal Health Service Utilization at Community Health Centre: A Mixed-Methods Study in Indonesia" supports an interesting objective. By identifying these barriers, more effective strategies can be developed to overcome them and increase women's adherence to prenatal and postnatal care.
However, this review presents some methodological limitations, which may have implications for the quality of the results.
As far as I can judge as a non-native English speaker, the language needs improvement on some grammatical inaccuracies, and the text needs to be revised.
As suggested below, some details might be improved to increase the number of manuscripts.
Abstract: Review the objective to agree on the abstract and introduction. It suggests improving methodological procedures, namely the type of study, sample size and data analysis.
Introduction:
The first paragraph relating to maternal mortality has a very similar structure to the second paragraph, meaning the information becomes redundant.
It would be important to include epidemiological data on maternal mortality in Indonesia, as well as explore the relationship between the lack of prenatal surveillance and maternal mortality.
I suggest that, albeit briefly, the authors explain how the prenatal care network is organized and what type of public services they have.
Line 40 -41 – "The factors include the mother's and the family's characteristics, family support, internal healthcare providers, and other social aspects. "Explore how these variables matter.
Line 42 – "Although the family is a relatively insignificant part of society…". I did not understand this statement.
Line 52: Do you intend to explore perspectives or insights? they are different things...
Line 52 -53 - The objective of the research is written in a confusing way and mixes several things….it should be restructured here and in the summary. confusing objective because it mixes several things…. It should be similar in the abstract and introduction.
Methodology:
The qualitative study must include contact procedures, selection of participants, formation of groups (are members of the same family or different families in each focus group?), icebreakers, data collection procedures, etc. The use of focus groups is based on a set of assumptions omitted in this description that authors must include.
The way qualitative data analysis was carried out should also be more detailed.
Only audio recordings were used to collect data from the focus groups. Why didn't they use video recordings?
Page 3, line 95: Why did they conduct an interview and not apply questionnaires since it was a quantitative study?
Page 3, line 103. "Participants for the FGDs were 103 purposively selected to participate by asking the CHC staff." Clarify that information, please.
It is important to include details about the form of bivariate analysis. Quantitative analysis can be deepened using logistic regression models to identify the most relevant factors in the use of health services.
Results
The authors refer to the type of statistical tests carried out and the generic significance values in the results. This information should be part of the methods, namely, in the data analysis subchapter.
In quantitative analysis, the data are all presented in detail, but only one of the variables was significant. We suggest that you combine all non-significant variables, mention that no statistically significant differences were observed as a whole, and present the variables with statistical significance in more detail, as only these will be the subject of discussion.
Page 5, lines 157 to 161 –Very long sentences that make reading difficult and where it makes no sense to put significance values since the tests did not reveal any differences.
Page 7 - In qualitative analysis, authors include complete and very long sentences, which makes the text exhaustive. We suggest only following the speeches that are important for the discussion.
Discussion
The discussion mentions conflicting findings regarding the impact of employment and residential status on maternal health service utilization. A deeper analysis could explore the reasons for these discrepancies, such as cultural differences, socioeconomic factors, or variations in healthcare systems.
While the study highlights the importance of government policies in promoting maternal health, a more detailed discussion of specific policy recommendations could be included. For example, the authors could suggest strategies to improve the flexibility of CHC service hours or provide additional support for working women.
The authors could elaborate on potential future research directions, such as investigating the impact of specific interventions to improve maternal health service utilization or exploring the role of technology in facilitating access to care.
Additionally, the authors could reiterate the importance of addressing the identified barriers to maternal health service utilization and the need for further research to inform evidence-based interventions. Addressing these areas can enhance the discussion section, providing a more comprehensive and impactful interpretation of the study findings.
In the discussion, the authors must articulate the results obtained with the day methodologies more appropriately.
The argument should focus on the advantages of using public services. Given that women receive prenatal care even though they do it privately.
Conclusions
The authors can suggest some strategies to solve the identified problems (e.g., the importance of flexible working hours and adequate staffing in CHCs to accommodate the needs of working women and women from different backgrounds), as well as strategies to minimize the impact of cultural and social norms on women's decision-making processes.
References
Around half of the references used are more than five years old, and we recommend updating some of them.
Comments on the Quality of English LanguageDear Editor
This article presents some methodological limitations, which may have implications for the global quality.
The language needs significative improvement on some grammatical inaccuracies, and the text needs to be revised.
Author Response
Reviewer 1
Dear Reviewer,
Thank you for your thorough and constructive feedback on our manuscript, "Understanding Interpersonal Influences on Maternal Health Service Utilization at Community Health Centre: A Mixed-Methods Study in Indonesia." We appreciate your detailed suggestions and have addressed them to improve the quality and clarity of our manuscript. Below is our response to your comments:
|
No |
Comments |
Responses |
|
1 |
Abstract: Review the objective to agree on the abstract and introduction. It suggests improving methodological procedures, namely the type of study, sample size and data analysis. |
The objective has been revised to align with the introduction, clearly stating the aim to assess sociodemographic characteristics and explore family members' support for maternal health service utilization.
The methodological description now specifies the study design (explanatory sequential mixed methods), sample size (432 respondents), and data analysis techniques (bivariate analysis and conventional content analysis). |
|
2 |
Introduction: The first paragraph relating to maternal mortality has a very similar structure to the second paragraph, meaning the information becomes redundant. It would be important to include epidemiological data on maternal mortality in Indonesia, as well as explore the relationship between the lack of prenatal surveillance and maternal mortality. I suggest that, albeit briefly, the authors explain how the prenatal care network is organized and what type of public services they have. |
Redundancy in Maternal Mortality Information: The first and second paragraphs have been restructured to avoid redundancy while emphasizing the global and national importance of maternal health.
Epidemiological Data on Maternal Mortality in Indonesia: We have included recent data on maternal mortality rates in Indonesia and explored the relationship between inadequate prenatal surveillance and maternal mortality.
Prenatal Care Network in Indonesia: A brief explanation of the prenatal care network, including the services provided by CHCs and their role in the healthcare system, has been added. |
|
3 |
Introduction: Line 40 -41 – "The factors include the mother's and the family's characteristics, family support, internal healthcare providers, and other social aspects. "Explore how these variables matter.
Line 42 – "Although the family is a relatively insignificant part of society…". I did not understand this statement.
Line 52: Do you intend to explore perspectives or insights? they are different things...
Line 52 -53 - The objective of the research is written in a confusing way and mixes several things….it should be restructured here and in the summary. confusing objective because it mixes several things…. It should be similar in the abstract and introduction. |
Clarifications on Variables (Line 40-41): We expanded on how characteristics such as family support, healthcare providers, and social factors influence maternal health service utilization.
Unclear Statement (Line 42): The statement "Although the family is a relatively insignificant part of society..." has been revised to clarify the central role of family in supporting women’s health decisions.
Objective Clarity (Line 52-53: The objective has been rewritten to clearly distinguish between exploring perspectives and assessing insights. The revised objective is now consistent in both the abstract and the introduction. |
|
4 |
Methodology: The qualitative study must include contact procedures, selection of participants, formation of groups (are members of the same family or different families in each focus group?), icebreakers, data collection procedures, etc. The use of focus groups is based on a set of assumptions omitted in this description that authors must include. The way qualitative data analysis was carried out should also be more detailed. Only audio recordings were used to collect data from the focus groups. Why didn't they use video recordings? |
Details on Qualitative Study: Additional details have been provided, including (a) Contact procedures and participant selection (e.g., purposive selection with assistance from CHC staff). (b) Group composition (involving members from the same or different families). (c) Other data collection procedures used during FGDs.
Focus Group Assumptions: The assumptions underlying the use of FGDs have been stated, emphasizing their relevance in exploring interpersonal factors.
Qualitative Data Analysis: We elaborated on the steps of qualitative data analysis, including transcription, coding, and the application of conventional content analysis. Use of Audio Recordings: The rationale for using audio recordings instead of video recordings (e.g., cultural and logistical considerations) has been clarified. |
|
5 |
Methodology: Page 3, line 95: Why did they conduct an interview and not apply questionnaires since it was a quantitative study?
Page 3, line 103. "Participants for the FGDs were 103 purposively selected to participate by asking the CHC staff." Clarify that information, please.
It is important to include details about the form of bivariate analysis. Quantitative analysis can be deepened using logistic regression models to identify the most relevant factors in the use of health services. |
Quantitative Interviews vs. Questionnaires: The decision to use interviews instead of questionnaires is explained as a means to capture in-depth information on maternal health-seeking behaviors.
Participant Selection (Page 3, Line 103): We clarified that participants were purposively selected with the help of CHC staff to ensure representation from different family contexts.
Bivariate Analysis and Logistic Regression: Details about the bivariate analysis using SPSS have been expanded. Additionally, we discussed the potential for logistic regression in future studies to identify the most relevant factors. |
|
6 |
Results: The authors refer to the type of statistical tests carried out and the generic significance values in the results. This information should be part of the methods, namely, in the data analysis subchapter.
In quantitative analysis, the data are all presented in detail, but only one of the variables was significant. We suggest that you combine all non-significant variables, mention that no statistically significant differences were observed as a whole, and present the variables with statistical significance in more detail, as only these will be the subject of discussion. |
Statistical Test Details: Information about the statistical tests (e.g., Chi-square) and significance values has been moved to the Methods section.
Presentation of Non-Significant Variables: Non-significant variables have been combined, and it is noted that no statistically significant differences were observed as a whole. Variables with statistical significance are presented in more detail. |
|
7 |
Results: Page 5, lines 157 to 161 –Very long sentences that make reading difficult and where it makes no sense to put significance values since the tests did not reveal any differences.
Page 7 - In qualitative analysis, authors include complete and very long sentences, which makes the text exhaustive. We suggest only following the speeches that are important for the discussion. |
Long Sentences (Page 5, Lines 157-161): These sentences have been revised for clarity and conciseness, and the use of significance values has been appropriately adjusted. Qualitative Analysis (Page 7): Long excerpts from qualitative data have been shortened to focus on key findings relevant to the discussion. |
|
8 |
Discussion: The discussion mentions conflicting findings regarding the impact of employment and residential status on maternal health service utilization. A deeper analysis could explore the reasons for these discrepancies, such as cultural differences, socioeconomic factors, or variations in healthcare systems.
While the study highlights the importance of government policies in promoting maternal health, a more detailed discussion of specific policy recommendations could be included. For example, the authors could suggest strategies to improve the flexibility of CHC service hours or provide additional support for working women.
The authors could elaborate on potential future research directions, such as investigating the impact of specific interventions to improve maternal health service utilization or exploring the role of technology in facilitating access to care.
Additionally, the authors could reiterate the importance of addressing the identified barriers to maternal health service utilization and the need for further research to inform evidence-based interventions. Addressing these areas can enhance the discussion section, providing a more comprehensive and impactful interpretation of the study findings.
In the discussion, the authors must articulate the results obtained with the day methodologies more appropriately.
The argument should focus on the advantages of using public services. Given that women receive prenatal care even though they do it privately. |
Conflicting Findings on Employment and Residential Status: We have expanded the analysis to explore reasons for discrepancies, considering factors such as cultural differences, socioeconomic conditions, and healthcare system variations.
Policy Recommendations: Specific policy recommendations have been added, including extending CHC service hours and providing additional support for working women to improve maternal health service utilization.
Future Research Directions: Potential directions for future research are elaborated, such as investigating the impact of specific interventions and exploring the role of technology in facilitating access to care.
Barriers and Evidence-Based Interventions: We emphasized the importance of addressing barriers to maternal health service utilization and the need for further research to develop evidence-based interventions.
Focus on Public Services: We have revised the discussion to highlight the advantages of public services while acknowledging the preference for private care in certain contexts. |
|
9 |
Conclusions The authors can suggest some strategies to solve the identified problems (e.g., the importance of flexible working hours and adequate staffing in CHCs to accommodate the needs of working women and women from different backgrounds), as well as strategies to minimize the impact of cultural and social norms on women's decision-making processes. |
Strategies to Address Barriers: We have suggested specific strategies, such as: a) Extending CHC service hours to accommodate working women. b) Improving staffing levels at CHCs. c) Addressing cultural and social norms that influence decision-making processes. |
|
10 |
References Around half of the references used are more than five years old, and we recommend updating some of them. |
Approximately half of the references have been updated to ensure they are more recent (within the past five years). |
We are grateful for the reviewer’s valuable feedback, which has significantly improved our manuscript. We hope that the revisions meet your expectations, and we look forward to your further comments.
Thank you.
Sincerely,
Herwansyah (On behalf of the authors)

Reviewer 2 Report
Comments and Suggestions for Authors
This study investigates the utilization of maternal health services at community health centers in Indonesia, focusing on the role of interpersonal and family support. It is appreciable that a mixed-methods approach, combining quantitative and qualitative data, is used. However,
1. The title can be "community health centers" instead of "community health centre."
2. Specific aspects of interpersonal influence can be mentioned in the introduction. The research objectives needed to be clearly defined.
3. The research employs an explanatory sequential mixed-methods design, which is well-detailed. Still, the instruments used in the survey could be described in more detail (e.g., types of questions, specific scales used). Also, about the validity and reliability of tools needed to be mentioned.
4. The results are presented clearly. Add a concise summary at the end of each main section (quantitative and qualitative) to make more clarity. Highlight if there are any connections between quantitative and qualitative findings as data triangulation.
5. Acknowledge that generalizability to other regions in Indonesia or similar settings may be limited in limitation of the research.
6. Include future research suggestions at the end of the discussion.
Comments on the Quality of English LanguageThe quality of the English language in the study is generally good and clear, but some areas could benefit from refinement for better readability and precision.
Author Response
Reviewer 2
Dear Reviewer,
Thank you for your thorough and constructive feedback on our manuscript, "Understanding Interpersonal Influences on Maternal Health Service Utilization at Community Health Centre: A Mixed-Methods Study in Indonesia." We appreciate your detailed suggestions and have addressed them to improve the quality and clarity of our manuscript. Below is our response to your comments:
|
No |
Comments |
Responses |
|
1 |
The title can be "community health centers" instead of "community health centre." |
The title has been updated to "Understanding Interpersonal Influences on Maternal Health Service Utilization at Community Health Centers: A Mixed-Methods Study in Indonesia" to reflect your suggestion. |
|
2 |
Specific aspects of interpersonal influence can be mentioned in the introduction. The research objectives needed to be clearly defined. |
a) Specific Aspects of Interpersonal Influence: Additional details on specific aspects of interpersonal influence, such as family encouragement, spousal support, and emotional support, have been incorporated into the introduction. b) Research Objectives: The research objectives have been rewritten to explicitly define the focus on exploring interpersonal and family support factors and their impact on maternal health service utilization. |
|
3 |
The research employs an explanatory sequential mixed-methods design, which is well-detailed. Still, the instruments used in the survey could be described in more detail (e.g., types of questions, specific scales used). Also, about the validity and reliability of tools needed to be mentioned. |
a) Survey Instruments: A detailed description of the survey instruments has been added, including the types of questions (e.g., multiple-choice, Likert scales) and specific constructs measured (e.g., family support, healthcare access). b) Validity and Reliability: Information about the validity and reliability of the tools has been included. We describe the pilot testing conducted with 30 participants to refine the survey and ensure reliability, as well as the use of validated frameworks for the qualitative coding. c) Explanatory Sequential Design: While the design was well-detailed, we have clarified the integration process between the quantitative and qualitative phases, specifically how findings from the survey informed the focus group discussion (FGD) questions. |
|
4 |
The results are presented clearly. Add a concise summary at the end of each main section (quantitative and qualitative) to make more clarity. Highlight if there are any connections between quantitative and qualitative findings as data triangulation. |
At the end of each results section (quantitative and qualitative), a brief summary has been added to enhance clarity. These summaries highlight key findings and their relevance. Connections between quantitative and qualitative findings have been explicitly discussed to demonstrate how the two data sources complement each other and provide a more holistic understanding of the phenomena studied. |
|
5 |
Acknowledge that generalizability to other regions in Indonesia or similar settings may be limited in limitation of the research. Include future research suggestions at the end of the discussion. |
Acknowledgment has been added regarding the limited generalizability of the findings to other regions in Indonesia or similar settings due to the specific cultural and demographic contexts of the study sites. |
We are grateful for the reviewer’s valuable feedback, which has significantly improved our manuscript. We hope that the revisions meet your expectations, and we look forward to your further comments.
Thank you.
Sincerely,
Herwansyah (On behalf of the authors)

Round 2
Reviewer 2 Report
Comments and Suggestions for Authors
Revise the title to community health centers.
Align the table properly and adhere to the font.
Comments on the Quality of English Languagebetter to be edited.